# The association between guidelines adherence and clinical outcomes during pregnancy in a cohort of women with cardiac co-morbidities

Sandra Millington[1]*, Suzanne Edwards[2], Robyn A. Clark[3], Gustaaf A. Dekker[4], Margaret Arstall[5]

1 Adelaide Medical School, University of Adelaide, Adelaide, South Australia, Australia, 2 Adelaide Health Technology Assessment (AHTA), School of Public Health, University of Adelaide, Adelaide, South Australia, Australia, 3 College of Nursing and Health Science, Flinders University, Adelaide, South Australia, Australia, 4 Women's and Children's Division Northern Adelaide Health Local Network (NAHLN) and Obstetrics and Gynaecology, University of Adelaide, Adelaide, South Australia, Australia, 5 Cardiology Unit NAHLN, University of Adelaide, Adelaide, South Australia, Australia

* sindy.millington@adelaide.edu.au

**Data Availability Statement:** All relevant data are within the manuscript and its Supporting Information files.

## Abstract

### Background/Aims

Maternal and infant morbidities associated with pregnant women with cardiac conditions are a global issue contingent upon appropriate care. This study aimed to describe the clinical variables and their association with the adherence scores to perinatal guidelines for pregnant women with cardiac conditions. The clinical variables included cardiac, perinatal, and neonatal outcomes and complications.

### Methods

Using a retrospective cross-sectional medical record audit, data were abstracted and categorised as cardiac, obstetric, and neonatal predictors. Linear regression modelling was used to find the mean difference (MD) in adherence scores for each predictor, including a 95% confidence interval (CI) and a significance value for all the three categories' clinical outcomes.

### Results

This maternal cohort's (n = 261) cardiac complications were primarily arrhythmias requiring treatment (29.9%), particularly SVT (28%), a new diagnosis of valvular heart disease and congenital heart disease (24%) and decompensated heart failure (HF) (16%). Women with HF had associated increased adherence scores (MD = 3.546, 95% CI: 1.689, 5.403) compared to those without HF. Elective LSCS mode of delivery was associated with a higher adherence score (MD = 5.197, 95% CI: 3.584, 6.811) than non-elective LSCS subgroups. Babies admitted to intensive /special care had greater adherence to the guidelines (MD = 3.581, 95% CI: 1.822, 5.340) than those not requiring the same care.

**Funding:** The author(s) received no specific funding for this work.

**Competing interests:** The authors have declared that no competing interests exist.

## Conclusions

Some pregnancy associated complications and morbidities were associated with higher adherence scores, reflecting that a diagnosis, identification of morbidities or risk factors, initiation of appropriate multidisciplinary involvement and adherence to guidelines were associated. Conversely, potentially avoidable major complications such as sepsis were associated with a low adherence score.

## Trial registration

ACTRN12617000417381.

## Introduction

Cardiovascular disease is a major cause of maternal mortality, accounting for up to 15% of maternal deaths and complicating between 1% and 4% of pregnancies, with the associated morbidities becoming an increasingly recognised issue, warranting our attention [1–4]. However, limited data is available on cardiac disease in pregnancy in Australia, and research is underreported [5, 6]. The national maternal mortality ratio, i.e. the number of deaths per 100, 000 women giving birth, was reportedly below ten from 2008 to 2017 [6, 7]. Yet cardiac-related maternal deaths occurring during pregnancy are classified as indirect maternal deaths due to preexistent medical conditions or disease acquired during pregnancy rather than the consequence of direct obstetric death [5]. Heart disease in pregnancy consists of a heterogeneous group of disorders inclusive of corrected and uncorrected congenital lesions, valvular disease, arrhythmias, aortopathies, cardiomyopathies, acquired atherosclerotic heart disease, genetic conditions, and cardiac complications associated with severe hypertensive disorders [6, 8]. Due to the increased physiological stress of pregnancy on the maternal cardiovascular system, maternal preexisting or cardiac pathological conditions acquired during pregnancy may have different outcomes than those in the non-pregnant state [9].

## Background and rationale

The two broad categories of cardiac conditions in pregnancy are preexistent heart disorders and pregnancy-acquired cardiac conditions [10, 11]. Congenital heart disease (CHD) is the predominant diagnosis of all cardiac lesions in developed countries (75% to 82%). Conversely, rheumatic heart disease comprises 56% and 89% of pregnancy-related cardiac conditions reported in developing countries, i.e. middle and low-income, respectively [8]. However, acquired cardiac complications may occur throughout pregnancy, including six months postpartum, when women recover to their prepregnancy haemodynamic status.

Women with a preexistent medical condition or prior adverse pregnancy outcomes are at increased risk of cardiac complications in subsequent pregnancies and later in life [4, 12, 13]. Adverse pregnancy outcomes include complications, such as maternal placental syndromes and gestational diabetes mellitus (GDM). Likewise, pregnant women with cardiac conditions are at increased risk of obstetric and neonatal complications [14–19]. Therefore, high risk cardiac pregnancies present clinicians with significant challenges.

Consequently, a comprehensive antenatal assessment and, ideally, preconception counselling are firmly recommended in best practice [20]. A healthy mother and baby are the aspirations of all pregnant women and the health care team delivering care during their pregnancy.

The clinical outcomes for women with cardiac conditions during pregnancy are influenced by the health care 'providers' expertise and knowledge of best practice perinatal guidelines and the healthcare system's capacity or limitations [20].

During pregnancy, cardiovascular disease (CVD) management guidelines have been informed by knowledge generated through case reports, registries, and expert opinion rather than evidence from randomised clinical trials [2, 5, 21]. In both South Australia and Western Australia, statewide guidelines integrate all levels of evidence to aspire to reduce maternal mortality and morbidity [22, 23]. This present study contributes to the broader research, of established statewide cardiac-specific perinatal guidelines for women with cardiac disease during pregnancy across the three major public hospitals, in Adelaide, South Australia. The aim was to measure clinical practice against guideline recommendations to verify real-life practice and identify gaps for quality improvement measures.

## Aim

The study aimed to describe clinical variables, i.e., morbidities, outcomes, interventions, and their association with the adherence score for the cardiac-specific statewide perinatal guidelines for women with cardiac conditions during pregnancy, with the following objectives:

- Describe the frequency of clinical morbidities, outcomes, and interventions in the preexistent and acquired cardiac groups of the cohort of women with cardiac conditions during pregnancy.

- Determine the association between the adherence to guidelines score and clinical variables, i.e., outcomes, complications and required interventions.

## Methods

### Study design and population

This research is a preplanned sub-study of the retrospective cross-sectional observational audit that evaluated adherence to clinical practice guidelines for South Australian pregnant women with cardiac conditions between 2003 and 2013. Details of the study design, data abstraction tool, settings, selection criteria and the evaluation of adherence to clinical practice guidelines across three hospital sites were published previously. In summary, we showed that variance in adherence correlated with the exposure to the higher acuity cases and appropriate up-referral to the only high risk referral centre for the state, the quartenary hospital two, which provided (level six) care, i.e. maternal cardiac and neonatal intensive care services. Likewise, hospital one offered intermediate care (level five) inclusive of tertiary maternal services, intensive care and specialised neonatal care services but excluded babies less than 32 weeks. Also, hospital three was a stand-alone maternity unit attached to a large neonatal intensive care, yet without a maternity intensive care service onsite [20].

The previous study evaluated the adherence to recommendations within the guidelines. The predictors for adherence were examined, such as the hospital sites, the two broad cardiac groups, underlying cardiac causes, risk level of pregnancy, multidisciplinary team collaboration, maternal and baby demographic factors. The participants included only women with 'preexistent' and 'newly acquired' heart disease during pregnancy, with a more comprehensive discussion on the selection criteria presented in Millington (2020) et al. [20].

The cohort's broad spectrum of maternal cardiac diagnosis and neonatal diagnosis which includes rare syndromes namely Scimitar, Goldenhar and Noonan are available in S1 and S2 Tables [24–26].

## Ethics approval

South Australian Health Human Research Ethics Committee approved this study (Reference HREC 13 TQEH/LMH 226: Extension to Approval 03/08/2015). The Research Ethics approval granted the researcher access to medical records to collect data, and consent was not required from the participants. Anonymity was ensured with the 'participants' identifiable information coded and stored with a separate password-protected file on the university server and labelled for deletion in 15 years.

## Data sources /measurements

**Guideline adherence score.**  The total adherence score, a numerical value for each participant, reflected the adherence to the contents of the statewide perinatal guidelines. Scoring encompassed preconception counselling, antenatal care (clinical assessment and investigations), foetal wellbeing assessments, planned risk stratification, mode of delivery and labour recommendations as per guidelines [20]. A scoring system was devised that measured adherence variables used in this study were equally weighted, giving a maximum score of forty to measure adherence. Positive documentation of the guidelines, regardless of the entry point, achieved a score. A minimum score of acceptable guideline adherence was determined after comparing the two cardiac groups mean, and median adherence scores with expert review of selected cases identifying the minimum expected care. From this analysis, a score of 35 (for the 'preexistent') and 17 (for the 'newly acquired') cardiac conditions were deemed acceptable guideline adherence.

**Clinical outcomes.**  The data collected from the retrospective clinical audit of medical records identified pregnancy complications for mother and baby, including morbidities, outcomes, the expected clinical investigations and interventions. Therefore, the string data required coding and categorisation. Likewise, variables utilised in the statistical analysis were categorised into three groups, namely: maternal cardiac (n = 17), obstetric (n = 17) and neonatal (n = 20) see S3 Table. Explicitly, the clinical variables for both primary and secondary clinical outcomes, investigations, and interventions during pregnancy could alter the trajectory for mother and baby, thus influencing the uptake of guidelines and acute complications related to poor guideline adherence [19].

## Data cleaning and screening process

The study implemented a three-step process to screen and clean data, repetitive screening, diagnosis and editing [27]. The procedures encompassed checking for consistencies in the data flow, exploring construing factors, comparing the analysis dataset and raw database for errors such as double entry, missing values, unreadable data, and writing error. Those cases with data missing for the clinical variables were excluded in the final analysis. Thus, the sample size reduced from 271 to 261 for the final analysis.

## Bias

Inherent bias may be apparent when women presented late to prenatal care or delivered prematurely, such that their access to guideline-based healthcare was delayed or curtailed. Furthermore, data collection was by a single reviewer (author one) with selected case adjudication sessions undertaken with authors four and five who are specialists in a tertiary hospital multidisciplinary team for high risk pregnancies.

## Sample size and power calculations

A sample size and power calculation were performed and reported on in a previous paper on this topic [20] based on adherence to guidelines. The sample size acquired provides a substantial 'margin for error' to compensate for missing data, but possibly not for rare events such as maternal deaths (n = 1).

## Clinical outcome variables

The clinical outcomes have been subdivided into three groups, namely cardiac, obstetric and neonatal.

**Cardiac variables.** The cardiac data collected were adverse cardiac events in the puerperium, including six months post-partum. Our primary clinical outcomes were maternal cardiac death, cardiac arrest, decompensated heart failure (HF) and acute pulmonary oedema, acute myocardial infarction (AMI), and a new diagnosis of valvular heart disease (VHD) or congenital heart disease (CHD) diagnosis during pregnancy [3, 28–31]. Women in the preexistent cardiac group diagnosed with additional cardiac pathologies such as VHD or CHD to their known heart disease during pregnancy were recorded.

Secondary clinical outcomes were arrhythmias requiring treatment and non-specific chest pain. Likewise, all invasive cardiac interventions appropriate to the statewide guidelines were included. Non-invasive procedures included computed tomography (CT), pulmonary angiography (CTPA), and echocardiography. Invasive procedures included coronary angiography with or without percutaneous coronary intervention (PCI), electrophysiology (EP) studies with or without accessory pathway ablation, placement of a cardiac internal electrical device (CIED), balloon valvoplasty and cardiac surgery [28, 32, 33].

**Obstetric variables.** Given the known correlation between obstetric complications and pregnant women with cardiac conditions, obstetric data were collected [4, 13]. These included gestational hypertension (GH), preeclampsia (PET), antepartum and post-partum haemorrhage, placental dysfunction, i.e., placenta praevia and accreta, threatened premature onset of labour, preterm birth, failure to progress in labour, intrauterine growth restriction (IUGR) and maternal sepsis [4, 16]. Also, the guidelines recommend vaginal delivery for women with cardiac conditions, with a shortened active phase in the second stage of labour to mitigate cardiovascular stress and potential complications; therefore, data on delivery methods were collected [34].

Data were collected for normal vaginal deliveries (NVD), instrumental vaginal deliveries (forceps and vacuum device), both elective and emergency lower segment caesarean sections (LSCS) and where the surgical approaches proceeded to hysterectomy or laparotomy. Likewise, where the women's planned delivery deviated with potential changes in adherence to the perinatal guidelines was examined. The obstetric variable 'deviation from the planned delivery mode' captured this data.

**Neonatal variables.** Preexistent maternal heart disease, possibly via a decline in maternal cardiac output during pregnancy (IUGR) and obstetric complications, are associated with an increased risk of neonatal complications [14, 19]. Therefore, this study collected data for neonatal complications. Our primary outcomes of neonatal mortality (death within 30 days of delivery), intrauterine foetal death (IUFD) (> 20 weeks gestation) and medical termination of the pregnancy for a maternal cardiac condition. Secondary neonatal outcomes of prematurity (< 37 weeks), clinical diagnosis of IUGR, congenital heart disease and genetic abnormalities data [35]. Perinatal data had birthweight <10[th] percentile or birthweight <2500g if the gestational age was unknown, low Apgar scores (< 7 at 1 and 5 minutes), and respiratory distress syndrome (RDS) with and without respiratory support. The data collected for resuscitation

measures at birth ranged from low intensity, i.e., tactile stimuli, suction, oxygen therapy with bag-mask ventilation, invasive intubation, and ventilation. Data for interhospital retrievals with admissions to Neonatal Intensive Care Unit (NICU) or Special Baby Care Unit level 3 (SBCU3) were collected [29, 36, 37].

## Statistical analysis

Analyses were performed using SPSS for Windows (version 26.0 IBM Corp, Armonk, NY). Descriptive analyses of the overall adherence score, frequency tables of the relevant categorical data and coded variables for women in the two cardiac groups were carried out. Cross-tabulations of the categorical variables and cardiac group indicated whether these variables were related. The total adherence score was normally distributed on inspection of the histogram and presented as mean +/- standard deviation (SD).

Univariate linear regression analysis of the total adherence score versus the separate cardiac, obstetric, and neonatal categories was performed (see S5–S7 Tables). The following are reported for all clinical variables: mean difference with 95% confidence interval (CI), significance value, adjusted R Squared, change in R Squared, and frequencies. The linear regression assumptions: normality of residuals and homoscedasticity (equal spread of variance) were found to be upheld in all univariate models by inspection of histograms and scatterplots of residual and predicted values.

The authors performed multivariable regression analysis using a p value cut-off criterion of 0.2 on the univariate linear regression results [38, 39]. Specifically, the inclusion of all covariates with p value <0.2 on univariate regression analysis were included in an initial multivariable model, followed by backward elimination until all remaining covariates had a p value < 0.2 [39] refer to S5–S7 Tables. Multivariable regression modelling with backwards elimination was undertaken on three separate regression models for obstetrics, neonatal and cardiac predictors, respectively (refer to Tables 2–4). This is how the final multivariable parsimonious model (presented in Table 5) was built.

## Results

### Clinical characteristics

The cohort included 261 women of unmatched preexistent cardiac (n = 139) and acquired cardiac groups (n = 122). Updated baseline characteristics are reported in S4 Table. The mean adherence score in the preexistent cardiac group was 17.7 versus 15.2 in the acquired cardiac group. In the preexistent group, more women were from a rural location 30 (21.6%) and aboriginal ethnicity 21 (15%), compared with the acquired group 15 (12%) and 11 (9%). The descriptive frequencies of the categorical clinical variables for the overall cohort of women and the two cardiac groups are reported in Table 1.

**Clinical morbidities, outcomes, and interventions.**   The primary clinical outcome of mortality was low for the cohort (n = 1). The only maternal death occurred in the acquired cardiac group and was associated with a low adherence score of eight. There were no neonatal deaths; however, one intrauterine death late in pregnancy occurred in the acquired cardiac group. One medical termination of pregnancy occurred in the first trimester due to severe mitral valve disease in the preexistent cardiac group.

Within the overall cohort, 11 women experienced a cardiac arrest, predominantly in the acquired cardiac group. Similarly, HF occurred in 42 cases (16%) and was more frequent in the acquired cardiac group (22%) n = 27, compared with the preexistent cardiac group (11%) n = 15. There were 78 (29%) cases of sustained arrhythmias requiring treatment in the overall cohort, with (28%) n = 73 SVT and mainly in the acquired cardiac group (44%) n = 54. Non-

**Table 1. The descriptive statistics for clinical variables in the preexistent and acquired cardiac groups of women in South Australia during 2003 and 2013.**

| Cardiac clinical variables | Combined Cardiac Groups n (%) N = 261 | Preexisting cardiac (PEC) n (%) N = 139 | Acquired cardiac (AC) n (%) N = 122 |
|---|---|---|---|
| Maternal Cardiac Death | 1 (0.4) | 0 | 1 (0.8) |
| Cardiac Arrest | 11 (4.2) | 3 (2.2) | 8 (6.6) |
| Decompensated Heart Failure | 42 (16.1) | 15 (10.8) | 27 (22.1) |
| Acute Myocardial Infarction (AMI) | 5 (1.9) | 0 | 5 (4.1) |
| VHD/CHD diagnosis during pregnancy | 63 (24.1) | 48 (34.5) | 15 (12.3) |
| Sustained Arrhythmias + Tx | 78 (29.9) | 21 (15.1) | 57 (46.7) |
| Supraventricular Tachycardia (SVT) | 73 (28.0) | 19 (13.7) | 54 (44.3) |
| Bradyarrhythmias without syncope and HF | 12 (4.6) | 1 (0.7) | 11 (9.0) |
| Non-specific chest pain | 52 (19.9) | 18 (12.9) | 34 (27.9) |
| Cardiovascular Implantable Electrical Device (CIED) | 6 (2.3) | 1 (0.7) | 5 (4.1) |
| Cardiac surgery required | 6 (2.3) | 5 (3.6) | 1 (0.8) |
| Balloon Valvoplasty required | 3 (1.1) | 2 (1.4) | 1 (0.8) |
| Cardiogenic shock with Intra-Aortic Balloon Pump (IABP) | 1 (0.4) | 0 | 1 (0.8) |
| **Need for invasive cardiac investigations.** | | | |
| EP studies ±Cardiac ablation | 9 (3.4) | 1 (0.7) | 8 (6.6) |
| CTPA+ Echocardiogram investigations required | 26 (10) | 4 (2.9) | 22 (18.0) |
| Coronary angiogram ± PCI | 14 (5.4) | 1 (0.7) | 13 (10.7) |
| *Other Non-cardiac complications | 16 (6.2) | 12 (8.6) | 4 (3.3) |
| **Obstetric clinical variables** | **Combined cardiac Groups** n (%) | **Preexisting cardiac conditions (PEC)** n (%) | **Acquired cardiac conditions (AC)** n (%) |
| **Modes of delivery** | | | |
| Normal vaginal delivery (NVD) | 102 (39.1) | 55 (39.6) | 47 (38.5) |
| Assisted vaginal delivery (VD) | 27 (10.3) | 11 (7.9) | 16 (13.1) |
| Elective Lower Segment Caesarean Section (LSCS) | 74 (28.4) | 49 (35.3) | 25 (20.5) |
| Emergency LSCS | 54 (20.7) | 23 (16.5) | 31 (25.4) |
| Emergency LSCS plus Hysterectomy | 4 (1.5) | 1 (0.7) | 3 (2.5) |
| †Deviated from planned mode of delivery | 92 (35.2) | 42 (30.2) | 50 (41) |
| **Obstetric clinical morbidities/ interventions** | | | |
| Failure to progress in labour (FTP) | 28 (10.7) | 13 (9.4) | 15 (12.3) |
| Threatened premature labour (TPL) | 30 (11.5) | 16 (11.5) | 14 (11.5) |
| Placenta Accreta | 3 (1.1) | 2 (1.4) | 1 (0.8) |
| Placenta Praevia | 8 (3.1) | 3 (2.2) | 5 (4.1) |
| Antepartum Haemorrhage (APH) | 9 (3.4) | 3 (2.2) | 6 (4.9) |
| Postpartum haemorrhage (PPH) | 36 (13.8) | 20 (14.4) | 17 (13.1) |
| Pregnancy Induced Hypertension (PIH) | 39 (14.9) | 16 (11.5) | 23 (18.9) |
| Preeclampsia (PET) | 25 (9.6) | 12 (8.6) | 13 (10.7 |
| Suspected Intrauterine Growth Restriction (IUGR) | 10 (3.8) | 8 (5.8) | 2 (1.6) |
| Sepsis | 15 (5.7) | 6 (4.3) | 9 (7.4) |
| **Neonatal clinical variables** | **Combined cardiac groups,** n (%) | **Preexisting cardiac conditions (PEC),** n (%) | **Acquired cardiac conditions (AC),** n (%) |
| ‡Alive and well | 189 (72.4) | 101 (72.7) | 88 (72.1) |
| IUFD/stillbirth | 1 (0.4) | 0 | 1 (0.8) |
| TOP (Medical) | 1 (0.4) | 1 (0.7) | 0 |
| Prematurity | 46 (17.6) | 20 (14.4) | 26 (21.3)) |
| Prematurity + NICU/SBCU3 transfer | 42 (16.1) | 18 (12.9) | 24 (19.7) |

(*Continued*)

**Table 1.** (Continued)

| | | | |
|---|---|---|---|
| Active Resus +O$_2$/BMV/Intubate/IPPV | 34 (13) | 19 (13.7) | 15 (12.3) |
| Retrieval with CPAP ventilation | 9 (3.4) | 5 (3.6) | 4 (3.3) |
| Active Resus low invasive | 25 (9.6) | 9 (6.5) | 16 (13.1) |
| Septic workup/Sepsis | 18 (6.9) | 6 (4.3) | 12 (9.8) |
| Respiratory Distress syndrome (RDS) | 39 (14.9) | 23 (16.5) | 16 (13.1) |
| RDS required CPAP | 21 (8.0) | 13 (9.4) | 8 (6.6) |
| §SFD clinical diagnosis | 10 (3.8) | 8 (5.8) | 2 (1.6) |
| Diagnosis of Congenital Heart Disease (CHD) | 6 (2.3) | 5 (3.6) | 1 (0.8) |
| Diagnosis of Non-cardiac Congenital Abnormalities | 8 (3.1) | 6 (4.3) | 2 (1.6) |
| ‖Apgar Score < 7 at 1 min | 35 (13.4) | 17 (12.2) | 19 (15.4) |
| ‖Apgar Score < 7 at 5 min | 18 (6.9) | 9 (6.5) | 10 (8.0) |
| NICU admission | 73 (28) | 38 (27.3) | 35 (28.7) |
| SBCU3 admission | 29 (11.1) | 12 (8.6) | 17 (13.9) |

VHD/CHD: Valvular Heart Disease or Congenital Heart Disease Diagnosis, HF: Heart failure, CIED: Cardiovascular Implantable Electronic Device, e.g. Pacemaker / implantable cardioverter defibrillator, Arrhythmias + TX: Sustained Arrhythmias requiring treatment, EP studies/Cardiac Ablation: Electrophysiology studies and Catheter Ablation, CTPA+ Echocardiogram: CT computed-tomography pulmonary angiogram, PCI percutaneous coronary intervention. NICU: Neonatal Intensive Care Unit, SBCU3: Special Baby Care Unit level 3, Active Resus+ O$_2$/BMV/ intubate /IPPV: active resuscitation that required oxygen administration(O$_2$), bag-mask ventilation (BMV) but also required intubation (intubate) and mechanical ventilation mode of intermittent positive pressure ventilation (IPPV), CPAP: mode of mechanical ventilation Continuous positive pressure ventilation

*Other non-cardiac complications: hepatic encephalopathy and bleeding gastric varices, oesophageal sclerotherapy.

† Deviated from planned delivery mode: planned NVD required alternative (listed) mode of delivery or planned Elective LSCS was precipitous SVD or Emergency LSCS.

‡ documented as "Alive and Well' by clinicians at birth, IUFD: Intrauterine Foetal Death ≥ 20 weeks gestation, TOP: Medical Termination of pregnancy < 24 weeks.

§ Clinical Diagnosis of IUGR/SFD small for dates

‖Missing data totals = ** 2, ***1(1& 5 min respectively).

specific chest pain 34 (27%) was prevalent in the acquired group and included only 5 (4%) AMI incidents for the cohort. New diagnosis of CHD and VHD during pregnancy occurred more often in women with a preexistent cardiac condition (34%) in comparison to the acquired group (12%) (See Table 1). These women were diagnosed with different cardiac pathology such as VHD or CHD to their known heart disease during pregnancy. A modest number of cardiac interventions (n = 58, 22%) encompassed cardiac diagnostic investigations, i.e., CTPA with an echocardiogram to the more invasive balloon valvoplasty and cardiac surgery, mainly in the preexistent cardiac group (See Table 1).

Whilst there were 101 (39%) normal vaginal delivery, 92 women (35%) deviated from their planned birthing mode, particularly in the acquired cardiac group (41%) n = 50. The birthing options of elective (28%) n = 74 and emergency LSCS (20%) n = 54 were found to be more frequent than assisted vaginal delivery (10%), n = 27. Additional information on the frequencies of obstetric complications in the cohort is in Table 1.

There were 189 (72%) neonates born with normal Apgar scores (>7 at 1 and 5 minutes), while 34 (13%) babies required active resuscitation at birth, particularly in the preexistent cardiac group. The cohort included six sets of twins, one set of triplets, and all were admitted to NICU for prematurity. Seventy-three (28%) babies were admitted to NICU, and 29 babies (11%) transferred to SBCU3. Nine babies in the cohort required invasive respiratory support

**Table 2. Multivariable linear regression model results for the adherence score versus maternal cardiac clinical variables.**

| Clinical variables | Mean difference Adherence score (95% CI) | Standardized Coefficients Beta | P value | N (%) N = 261 |
|---|---|---|---|---|
| Cardiac clinical outcomes. | | | | |
| Decompensated Heart failure (HF) | 5.062 (2. 976,7.147) | 0.286 | <0.001 | 42 (16) |
| Non- specific chest pain | 1.846 (-0.348,4.039) | 0.113 | 0.099 | 52 (19.9) |
| Cardiac interventions. | | | | |
| *CTPA+ Echocardiogram | -3.388 (-6.308, -0.467) | -0.156 | 0.023 | 26 (10) |
| Cardiovascular Implantable Electrical Device (CIED) | 4.801 (-0190, 9.792) | 0.111 | 0.059 | 6 (2.3) |
| Cardiac Surgery required | 5. 962 (0.968, 10.956) | 0.137 | 0.019 | 6 (2.3) |
| Balloon Valvoplasty required. | 10.027 (3.177,16.876) | 0.164 | 0.004 | 3 (1.1) |
| Coronary angiogram with/without †PCI | -2.646 (-6.187, 0.896) | -0.092 | 0.142 | 14 (5.4) |
| Other factors | | | | |
| Acquired cardiac group | -2.264 (-3.843,-0.686) | -0.174 | 0.005 | 122 (46.7) |

Significance p value <0.2. All clinical variables were yes vs no.

*CTPA+ Echocardiogram = Computed-tomography pulmonary angiogram and cardiac echocardiogram.

†PCI = percutaneous coronary intervention.

and retrieval. More neonates (n = 23; 16%) in the preexistent cardiac group experienced RDS, whilst 13 babies with RDS (9%) required respiratory support. A small number of babies (n = 6) were diagnosed with congenital heart disease and abnormalities, particularly in the preexistent cardiac group (See S2 Table).

**The association between the guideline adherence score and clinical variables.** The initial multivariable regression analysis results for the separate groups of clinical outcomes are presented in Tables 2–4. However, the final multivariable results for the association between guideline adherence score and clinical variables for the cohort are shown in Table 5. The cardiac variable of HF had the statistically most robust association with an improved guideline adherence score (p < 0.001). Women with HF had a mean guideline adherence score of 3.546

**Table 3. Multivariable linear regression model results for the adherence score versus obstetric clinical variables.**

| Clinical variables | Mean difference Adherence score (95% CI) | Standardized Coefficients Beta | P value | n (%) N = 261 |
|---|---|---|---|---|
| Obstetric Complications | | | | |
| Preexistent cardiac group | 1.957 (0.552, 3.362) | 0.150 | 0.007 | 139 (53.2) |
| Pregnancy Induced Hypertension (PIH) | 2.156 (-0.170, 4.482) | 0.118 | 0.069 | 39 (14) |
| Preeclampsia (PET) | 1.954 (-0.979, 4.887) | 0.088 | 0.191 | 25 (9.6) |
| Placenta Previa (PP) | 4.802 (-0.489,8.754) | 0. 127 | 0.017 | 8 (3.1) |
| Threatened premature onset of Labour | 2.015 (-0.131, 4.160) | 0.099 | 0.066 | 30 (11.5) |
| Abnormal Intrauterine Growth restriction (IUGR) / Small for dates (SFD) | 4.964 (1.366,8.561) | 0.147 | 0.007 | 10 (3.8) |
| Modes of Delivery | | | | |
| Assisted vaginal delivery (VD) | 1.946 (-0.427,4.319) | 0.091 | 0.108 | 27 (10.3) |
| Lower segment Caesarean Section (LSCS) (Elective) | 6.410 (4.733,8.087) | 0.445 | <0.001 | 74 (28) |
| Emergency LSCS | 2.983 (0.003,1.025) | 0.186 | 0.003 | 54 (20.7) |
| *Deviated from planned delivery mode | 1.610 (-0.168, 3.052) | 0.118 | 0.029 | 92 (35.2) |

Significance p value <0.2. All clinical variables were yes vs no.

* Deviated from planned delivery mode; those women with planned NVD who required alternate listed modes or Elective LSCS was precipitous SVD or Emergency LSCS.

**Table 4. Multivariable linear regression model results for the adherence score versus neonatal clinical variables.**

| Clinical variables | Mean difference Adherence score (95% CI) | Standardised Coefficients Beta | P value | n (%) N = 261 |
|---|---|---|---|---|
| Preexistent cardiac group | 2.357 (0.883, 3.832) | 0.181 | 0.002 | 139 (53.2) |
| Prematurity +*NICU/SBCU3 | 2.063 (-0.226, 4.351) | 0.115 | 0.077 | 42 (16.1) |
| Retrieval/ †CPAP/ Ventilation | -3.525 (-8.349, 1.299) | -0.099 | 0.151 | 9 (3.4) |
| RDS required CPAP | -3.021 (-6.891, 0.849) | -0.126 | 0.125 | 21 (8) |
| Low Apgar Score<7 at 5 minutes | 3.841 (-0.170, 7.852) | 0.150 | 0.060 | 18 (6.9) |
| NICU Admission | 4.966 (2.990, 6.943) | 0.343 | <0.001 | 73 ((28) |
| Sepsis workup/ sepsis | -2.311 (-5.396, 0.773) | -0.090 | 0.141 | 18 (6.9) |
| Diagnosis of Congenital Abnormalities | 4.544 (0.010, 9.098) | 0.121 | 0.050 | 8 (3.1) |

Significance p value <0.2. All clinical variables were yes vs no.

*NICU: Neonatal Intensive Care Unit, SBCU3: Special Baby Care Unit Level 3

†CPAP: mode of ventilation Continuous Positive Pressure ventilation

‡ RDS: Respiratory Distress Syndrome + CPAP continuous positive airway pressure.

units higher than those who did not have heart failure (mean difference (unstandardised beta coefficient of linear regression equation) = 3.546, 95% CI: (1.689, 5.403). There was a statistically significant association between the adherence score and the following cardiac variables: women in the preexistent cardiac group (p = 0.004), those women who had balloon valvoplasty (p = 0.004), those who required cardiac surgery (p = 0.041), and the investigations of CTPA inclusive of an echocardiogram (p = 0.047), whilst adjusting for all other covariates in the multivariable model. There was no association with adherence score in women who experienced cardiac arrest in the final multivariable analysis.

Elective LSCS mode of delivery had the statistically most robust association with an improved guideline adherence score (p < 0.001). Those women who had an elective LSCS had a mean guideline adherence score 5.197 units more than those who did not (mean difference = 5.197, 95% CI: 3.584, 6.811). The statistically significant associations between the guideline adherence score and other obstetric variables were emergency LSCS (p = 0.0120), diagnosis of preeclampsia (p = 0.034) and placenta praevia complication (p = 0.011), whilst adjusting for all other covariates.

The only neonatal variable reporting a statistically significant association with the guideline adherence score was NICU admission adjusting for all covariates in the model (p <0.001). Those women with babies who went to NICU admission had a mean adherence score 3.581 units greater than those without babies in NICU admission (mean difference = 3.581, 95% CI: 1.822, 5.340). A small number of babies diagnosed with congenital cardiac conditions (n = 6) are included in NICU admission, with some requiring further paediatric care and follow up (refer to S2 Table).

## Discussion

The study aimed to compare clinical outcomes against guideline recommendations to verify real-life practice and identify gaps for quality improvement measures. Particularly to describe the frequencies of clinical morbidities, outcomes and interventions in the cohort of women with cardiac conditions during pregnancy and determine their association with the adherence score. We found that the mortality and morbidity findings of this study were consistent with international research. Pregnancy in women with cardiac conditions is associated with significant morbidity, yet maternal cardiac mortality is reportedly rare (our cohort, n = 1) [28]. Our cohort reflected similar findings to the Cardiac Disease in Pregnancy II (CAREPREG II) study

**Table 5. Multivariable linear regression model results: Clinical variables versus adherence score of the clinical practice guidelines for South Australian pregnant women with cardiac conditions between 2003 and 2013.**

| Clinical variables | Mean Difference in Adherence score (95% CI) | Standardised Coefficients Beta | P value | Frequency |
|---|---|---|---|---|
| | | | | N = 261 |
| **Obstetric complications.** | | | | |
| Preeclampsia (PET) | 2.573 (0.199, 4.948) | 0.117 | 0.034 | 25 |
| Placenta Previa (PP) | 4.900 (1.136, 8.663) | 0.130 | 0.011 | 8 |
| **Modes of Delivery** | | | | |
| Assisted vaginal delivery (VD) | 1.631 (-0.593, 3.856) | 0.076 | 0.150 | 27 |
| LSCS (Elective) | 5.197 (3.584, 6.811) | 0.360 | <0.001 | 74 |
| Emergency LSCS | 2.418 (0.529, 4.307) | 0.151 | 0.012 | 54 |
| *Deviation from planned delivery | 1.104 (0.276, 2.485) | 0.81 | 0.116 | 92 |
| **Neonatal complications.** | | | | |
| Retrieval/ CPAP/ Ventilation | -3.868 (-8.108, 0.372) | -0.109 | 0.074 | 9 |
| Prematurity +NICU/SBCU3 | 1.602 (-0.436, 3.641) | 0.090 | 0.123 | 42 |
| Respiratory Distress Syndrome required CPAP. | 3.077 (-6.543, 0.381) | -0.129 | 0.081 | 21 |
| NICU admission | 3.581 (1.822, 5.340) | 0.247 | <0.001 | 73 |
| Low Apgar score <7 at 5minutes | 2.939 (-0.628, 6.505) | 0.115 | 0.106 | 18** |
| **Covariates/Predictors** | Mean Difference in Adherence score (95% CI) | Standardised Coefficients Beta | P value | Frequency |
| | | | | N = 261 |
| **Neonatal complications continued** | | | | |
| Diagnosis of Congenital abnormalities | 3.273 (-0.813, 7.360) | 0.087 | 0.116 | 8 |
| Septic workup /Sepsis | -2.613 (-5.355, 0.128) | -0.102 | 0.062 | 18 |
| **Cardiac complications /interventions.** | | | | |
| Decompensated Heart failure (HF) | 3.546 (1.689, 5.403) | 0.199 | <0.001 | 42 |
| Required Cardiac Surgery | 4.642 (0.199, 9.085) | 0.107 | 0.041 | 6 |
| Balloon Valvoplasty procedure | 5.293 (-0.864, 11.450) | 0.087 | 0.004 | 3 |
| CTPA+ Echocardiogram | -2.272 (-4.513, 0.031) | -0.105 | 0.047 | 26 |
| **Other factors** | | | | |
| Pre-existent cardiac group | 2.010 (0.643, 3.377) | 0.154 | 0.004 | 139 |

Significance P < 0.2. All covariates /predictors were yes vs no. LSCS: lower segment

* Deviated from planned delivery mode, i.e., planned normal vaginal delivery but required alternative (listed) modes, or planned Elective but precipitously SVD or Emergency LSCS

** Missing data totals for 2 cases. CTPA+ Echocardiogram: CT pulmonary angiogram and cardiac echocardiogram.

that cardiac complications in pregnant women with heart disease are more frequently maternal arrhythmias (30%) and heart failure (16%) (Table 1) [28]. This comprehensive retrospective study supports the hypothesis of an association between the adherence score to the guidelines and some clinical variables indicating improved outcomes with greater adherence (Table 2). But the most significant was decompensated heart failure (p < 0.001) and elective LSCS (p <0.001), contingent upon the inclusion of multidisciplinary team collaboration in high risk pregnancies [20].

One maternal cardiac death occurred post-partum. The mother presented to the hospital's emergency department in septic shock due to infective endocarditis post-discharge, following the successful term delivery of a live baby six weeks earlier. She died during this admission despite intensive care treatment. On her initial hospital presentation, she arrived in established labour, was diagnosed, and treated for a urinary tract infection. She had not attended antenatal care and was a known current illicit drug user. Therefore, her adherence score was low. One

IUFD occurred during the second trimester. The woman made a telephone enquiry about the absence of foetal movement, but she delayed presentation to the hospital where the diagnosis of IUFD was made. These findings underscore the importance of factors such as nonattendance to antenatal care, non-compliance plus medical error in a known intravenous illicit drug user, late presentation to the hospital resulting in missed care, and low guideline adherence score described elsewhere [20].

Cardiac arrests during pregnancy frequently lead to mortalities and severe maternal morbidities [40, 41]. No reported mortality associated with the cardiac arrests occurred in this cohort, but one woman experienced severe maternal morbidities following a post-partum cardiac arrest (S8 Table). The woman's medical history was multiparity, chronic hypertension, GDM, high BMI, and antepartum diagnosis of Left Bundle Branch Block (LBBB) on ECG. Also, late confirmation of global left ventricular dysfunction consistent with a dilated cardiomyopathy on echocardiogram. This woman repeatedly presented to the emergency departments at different hospitals with chest pain complaints, dysponea and tachycardia. However, the diagnostic echocardiogram occurred following her admission and post-partum cardiac review. Initially, the woman was not on cardiac telemetry and was discovered unresponsive and in cardiac arrest for an unknown period by a medical officer. Following resuscitation, she required prolonged intensive care and hospital admission complicated by a secondary PPH and hypoxic brain injury. She received treatment for her cardiomyopathy, including CIED insertion and tracheostomy for long term respiratory support. Unfortunately, over time neurological recovery was insufficient; therefore, she was discharged to long term residential care [41]. This cardiac arrest may have been prevented if medical staff had promptly adhered to guidelines so that appropriate treatment could be offered for this life-threatening cardiac condition.

The strength of the linear association between the cardiac groups and adherence to the guidelines was more significant in the preexistent cardiac group (global p value = 0.004) compared to the acquired cardiac group reiterating previous findings [20]. Women in the preexistent group had a mean adherence score of 2.010 units greater than those in the acquired cardiac group (mean difference = 2.010, 95% CI: 0.643, 3.374). The guidelines' efficacy and effectiveness were apparent when explanatory clinical variables were more strongly associated with higher adherence score and positive clinical outcomes [20]. Pregnant women with preexistent cardiac conditions, e.g. Tetralogy of Fallot (TOF), where multidisciplinary team management and a documented plan pertinent to the guidelines were in place, despite antepartum cardiac events typically experienced a good clinical course [20].

Women with pregnancy-acquired cardiac complications or congenital heart disease unmasked during pregnancy had a lower adherence score due to the change in their care trajectory during the antepartum period and experienced morbidities (See S1 Table). The lack of routine prenatal health (including cardiac) screening, preterm delivery, and late diagnosis in pregnancy contributed to a lack of awareness and preparedness previously reported [20, 42]. Clinicians were surprised and unprepared for potential cardiac complications in pregnancy if asymptomatic women in the first trimester later experienced sudden clinical deterioration. A study from Turkey demonstrated a 5.2% prevalence of cardiovascular disease (CVD) among asymptomatic first trimester women in a tertiary obstetric centre. This emphasises the importance of opportunistic as well as routine health and CVD screening by general practitioners in all patients whatever their age, and for newly pregnant women at their first antepartum visit [42]. Prenatal screening includes obtaining the pregnant woman's medical history and family members' medical history, cardiac status, and associated risk factors [20]. The authors recommend that skilled medical staff review the cardiovascular status of pregnant women with

associated risk factors, i.e. indigenous and refugee population for RHD, family cardiac history and known illicit drug user.

More women experienced sustained arrhythmias requiring treatment (30%) and decompensated heart failure (16%) in the acquired cardiac group compared to the preexistent cardiac group. Notably, in the final multivariable model, those pregnant women diagnosed with HF were associated with improved adherence to the perinatal guidelines reflected in the increased adherence score (global p <0.001). The clinicians' awareness of previous cardiomyopathies, early symptomatic onset in the antepartum period, and acute post-partum events, prompted adherence to the guidelines [15, 43, 44]. Antepartum HF was more likely to increase the adherence score as clinicians sought specific guideline recommendations to manage the heart failure appropriately during pregnancy [20]. With careful monitoring of volaemic status during pregnancy described in the perinatal guidelines, the heart failure regime ensured clinicians' 'preparedness' [33].

## Modes of delivery

The modes of delivery are an essential consideration for women with cardiac conditions to mitigate cardiovascular stress during the active phase of labour and yielded surprising results in this study. Although spontaneous or assisted vaginal delivery is preferred according to perinatal guidelines, the elective LSCS mode of delivery showed a statistically significant association with increased adherence to the guidelines adjusting for all other covariates with p value<0.001. This association demonstrates a consensus on the obstetric rationales for LSCS in women with cardiac conditions, medical conditions, and prior pregnancy complications who are more likely to experience obstetric and neonatal complications [14–19].

Women identified with adverse pregnancy outcomes such as malpresentation, premature onset of labour, multiple births, or clinical deterioration prompted elective LSCS to ensure the best result for mother and baby. As expected, the cardiac scenarios resulting in a decision for emergency or elective LSCS for solely cardiac indications were small (n = 20, 7.6%). Likewise, the rationale for elective LSCS included common obstetrical complications i.e., repeat LSCS (28%) placental dysfunction, i.e., placenta praevia and accreta (6%), pregnancy-induced hypertension, preeclampsia, and eclampsia (18%), delivery related, i.e., fail to progress (10%), malpresentation (2%) and baby-related indications such as compromised foetal wellbeing (4.9%) (S9 Table).

To understand these results, from a clinical perspective, elective LSCS occurred predominantly in the preexistent group (35%); clinicians had a low tolerance for significant obstetric complications. Often multiple adverse pregnancy complications influenced clinicians' decision to proceed with the surgical option of delivery. Women with significant preexistent cardiac conditions such as severe mitral valve disease and pulmonary hypertension, worsening decompensated heart failure and severe heart failure (NYHA classification IV) prompted early admission to hospital for cardiac optimisation. This included advanced life support in cardiac arrest, diuresis in decompensated heart failure, and respiratory support to manage hypoxia. Likewise, women required inotropic support for cardiogenic shock, haemodynamic monitoring during and immediately after delivery and management of cardiac arrhythmias with anti-arrhythmic medication, Direct Current (DC) cardioversion or implantation of an internal cardiac electrical device.

The rationale for elective LSCS included common obstetrical indications like placenta praevia, preeclampsia, or compromised foetal wellbeing. In our cohort, the high rate of elective LSCS (although not considered the first option in the statewide guidelines) increased guideline adherence scores related to other aspects of the guidelines yet improved the overall clinical

outcome for both mother and baby. Early inclusion of multidisciplinary team for surgical delivery options facilitated the uptake of guidelines recommendations such as assessing cardiac status. Therefore the results support the safety of elective LSCS in severe and unstable cardiac conditions. Emergency LSCS was more prevalent in the acquired cardiac group (25%) for obstetric reasons; however, emergency LSCS precipitated by cardiac events mid-trimester had low adherence score and poorer outcomes.

Admission to NICU was the only significant neonatal variable associated with increased adherence to guidelines. $P < 0.001$ indicating an appropriate response for the newborns' care in this cohort.

There were women with cardiac conditions who experienced unexpected non-cardiac complications during pregnancy. Two examples include intracranial bleeding treated with craniotomy and tumour in the distal trachea treated post-delivery with extracorporeal membrane oxygenation (ECMO) support for thoracotomy and debulking of tumour and insertion of a tracheal stent. All 'other' complications were included in the analysis; however, although clinically meaningful, none were significant.

## Conclusion

This study reassures clinicians that women in the preexistent cardiac group who had greater adherence to the perinatal guidelines achieved better clinical outcomes for both mother and baby, yet this did not guarantee an uneventful pregnancy journey. This result is encouraging despite the lack of Level 1A recommendations within these guidelines. There is insufficient robust published scientific and clinical evidence in this area of obstetric medicine. However, clinicians' awareness and preparedness of the women's preexisting cardiac conditions and the appropriate care process in the guidelines helped them anticipate potential pregnancy complications and prepare for interventions [20]. Likewise, pregnant women's health literacy has a significant role in guideline adherence; therefore, their cardiac conditions' health education is vital. In women with acquired cardiac conditions during pregnancy, often earlier pre-emptive warning 'signs' are insufficient. Therefore, adherence to cardiac disease guidelines begins at the cardiac event timeframe, resulting in a lower score. However, if the health care team and mother can rapidly readjust the care trajectory according to the guidelines, they may avoid severe complications. Therefore, adherence to the guidelines by both the pregnant women and the healthcare team improves clinical outcomes and provides clinicians with an opportunity for early diagnosis and interventions, mitigating potentially serious complications.

## Limitations

This study did not reflect state or national population proportions of women with cardiac conditions in pregnancy and encountered the anticipated limitations with retrospective medical record reviews, such as missing data and inaccurate documentation. The researcher examined individual case notes for additional documented evidence to confirm the care provided.

**Generalizability.** The data collected was limited to South Australia Health public hospitals due to access and data availability from medical records. Therefore, the sample size and findings do not reflect all women in South Australia.

## Supporting information

**S1 Table. Cardiac diagnosis.**
(PDF)

**S2 Table. Neonatal diagnosis of congenital heart disease and abnormalities.**
(PDF)

**S3 Table. Coding of clinical interventions, complications, and outcomes for SPSS variables for obstetric, neonatal and maternal cardiac.**
(PDF)

**S4 Table. Baseline characteristics for the cohort.**
(PDF)

**S5 Table. Univariate linear regression model results for adherence score versus obstetric clinical variables, neonatal clinical variables and maternal cardiac variables.**
(PDF)

**S6 Table. Univariate linear regression model results for adherence score versus obstetric clinical variables, neonatal clinical variables and maternal cardiac variables.**
(PDF)

**S7 Table. Univariate linear regression model results for adherence score versus obstetric clinical variables, neonatal clinical variables and maternal cardiac variables.**
(PDF)

**S8 Table. Cardiac arrests cases, associated pathologies and outcomes.**
(PDF)

**S9 Table. Cardiac, obstetric and other indications for emergency and elective lower segment caesarean section.**
(PDF)

## Acknowledgments

The authors acknowledge the invaluable support of the three sites' medical records departments.

Sandra Millington acknowledges support through an Australian Government Research Training Program scholarship. Robyn Clark acknowledges support through the Heart Foundation Future Leader Fellowship (APP100847).

## Author Contributions

**Conceptualization:** Sandra Millington.

**Data curation:** Sandra Millington.

**Formal analysis:** Sandra Millington, Suzanne Edwards.

**Funding acquisition:** Sandra Millington.

**Investigation:** Sandra Millington.

**Methodology:** Sandra Millington, Suzanne Edwards.

**Project administration:** Sandra Millington.

**Resources:** Sandra Millington.

**Software:** Sandra Millington.

**Supervision:** Suzanne Edwards, Robyn A. Clark, Gustaaf A. Dekker, Margaret Arstall.

**Validation:** Sandra Millington, Margaret Arstall.

**Visualization:** Sandra Millington, Robyn A. Clark, Gustaaf A. Dekker, Margaret Arstall.

**Writing – original draft:** Sandra Millington.

**Writing – review & editing:** Sandra Millington, Suzanne Edwards, Robyn A. Clark, Gustaaf A. Dekker, Margaret Arstall.

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
