## [Decision Letter · Decision Letter 0]

5 May 2021

PONE-D-21-08006

The association between guidelines adherence and clinical outcomes during pregnancy in a cohort of women with cardiac co-morbidities.

PLOS ONE

Dear Dr. Millington,

Thank you for submitting your manuscript to PLOS ONE. After careful consideration, we feel that it has merit but does not fully meet PLOS ONE’s publication criteria as it currently stands. Therefore, we invite you to submit a revised version of the manuscript that addresses the points raised during the review process.

We look forward to receiving your revised manuscript.

Kind regards,

Sara Ornaghi, M.D., Ph.D.

Academic Editor

PLOS ONE

Journal Requirements:

Reviewers' comments:

Reviewer's Responses to Questions

**Comments to the Author**

1. Is the manuscript technically sound, and do the data support the conclusions?

Reviewer #1: Yes

Reviewer #2: Partly

Reviewer #3: Yes

2. Has the statistical analysis been performed appropriately and rigorously? 

Reviewer #1: I Don't Know

Reviewer #2: N/A

Reviewer #3: Yes

3. Have the authors made all data underlying the findings in their manuscript fully available?

Reviewer #1: Yes

Reviewer #2: No

Reviewer #3: Yes

4. Is the manuscript presented in an intelligible fashion and written in standard English?

Reviewer #1: Yes

Reviewer #2: Yes

Reviewer #3: Yes

5. Review Comments to the Author

Reviewer #1: In my opinion, this is an interesting article because heart disease in pregnancy is an area of little available evidence. Currently, it has a lot of relevance since it is becoming a significant cause of maternal morbidity and mortality in both developed and developing countries. Despite this relevance and international guidelines that highlight the subject, most of the topics are recommendations made by experts who have little available evidence.

This type of study helps to support the recommendations of the guidelines. It is especially relevant to highlight the role of multidisciplinary groups, the early detection of heart disease in pregnant women, and the timely direction to the best care centers where it is possible to have a better follow-up of the guidelines.

Also, it is interesting to see that some pathologies, in particular, alert the health team, and they allow improving the standards of care. But, it is also a challenge to bring these levels of adherence to all patients, even when the diseases do not seem to be so complex.

This kind of article shows the work to be done and can raise awareness to achieve better results.

However, I consider it pertinent to review the following aspects:

1. To understand this article better, it is necessary to thoroughly review the study where the authors evaluated adherence to the guidelines. However, It could be helpful to add a brief description of the design aspects and the previous publication to permit the readers to inquire about the topic.

2. I consider it is essential to highlight the difference between the three hospitals mentioned.

3. Although maternal mortality was very low, the number of patients who presented with cardiac arrest was significant, so it would be relevant to report which were the pathologies that were associated with cardiac arrest in the description of outcomes (it could be a table)

4. Cesarean section deliveries were very high. It would be interesting to know the indication for cesarean section (cardiac vs obstetric) since most heart diseases can end by vaginal delivery, leaving a cesarean section for a cardiac indication to very selected cases.

It would be essential to know more characteristics that help define why the high rate of cesarean sections. The authors mention that state practices do not suggest elective cesarean section as the first option (and neither do the other pregnancy and heart disease guidelines) but, they recommend that this enhanced the results for both: mother and baby. So, it is essential to know in which scenarios the cesarean section was performed because if it was in severe and unstable conditions it would be justified.

5. 30-40% of patients had deviation in the delivery plan. As far as possible, given the retrospective nature, it would be good to know why the change occurred (change in the clinical condition that caused an indicated change or preference of the treating physician), as this has implications for quality of care and the impact of the concept of the multidisciplinary group.

6. The acronyms should be reviewed as some do not correspond or are not referenced before their use.

Reviewer #2: This was a sub-study of a retrospective cross sectional observational audit of 261 women that evaluated adherence to clinical practice guidelines for South Australian pregnant women between 2003-2012 previously published in PloS one in 2020. Although an interesting topic, it was difficult to understand the exact aims and hypothesis of this specific paper. Due to my confusion with the aims of this current paper, I reviewed the original submission which was clinically very useful as it looked at the assessment and the relationship to outcome as well as barriers to care. Although the results were reported as adherence versus non-adherence the tables and methods did not really reflect this.

Results were reported for two groups: Those with pre-existent cardiac issues and those with acquired cardiac issues. Table 1 shows multiple frequencies of cardiac outcomes, multiple obstetric clinical outcomes, and multiple neonatal clinical outcomes for the two different cardiac groups and the two groups combined. I believe the hypothesis was the association between adherence to guidelines and outcomes, but if that was the case it would be important to truly show characteristics between the adherent and non-adherent groups prior to showing the regression.

Multiple clinical outcomes were presented: Cardiac variables, obstetric variables and neonatal variables.

Specific questions include:

1) Was there data on congenital neonatal conditions, in particular neonatal heart disease and how did that relate to the outcomes?

2) What was included in the multivariate model?

3) Was there a power analysis done for the outcomes of interest, many were rare events

4) Most of the presentation of results seems to be a comparison of the outcomes in the preexistant versus

Reviewer #3: SUMMARY

Pregnant women with cardiac conditions are at increased risk of maternal, obstetric and neonatal complications. The two broad categories of cardiac conditions in pregnancy are preexistent heart disorders and pregnancy-acquired cardiac conditions.

In both South Australia and Western Australia, statewide guidelines integrate all levels of evidence to aspire to reduce maternal mortality and morbidity.

The same research group demonstrated in 2020 (1) that the adherence to the statewide guidelines developed and available in South Australia since 2010 is suboptimal across three SA Health public metropolitan, university-affiliated hospitals and variance in the level of adherence across the three hospitals correlated with the exposure to higher acuity cases.

Aim of the present sub-study is to describe the frequency of clinical morbidities, outcomes, and interventions in the preexistent and acquired cardiac groups of women with cardiac conditions during pregnancy and to determine the association between the adherence score to the guidelines and clinical variables.

Clinical outcome variables are divided in cardiac, obstetric and neonatal.

Descriptive frequencies of the categorical clinical variables for the overall cohort of women and

the two cardiac groups are reported.

Primary cardiac outcomes (maternal cardiac death, cardiac arrest, decompensated heart failure (HF) and acute pulmonary oedema, acute myocardial infarction (AMI), and a new diagnosis of valvular heart disease (VHD) or congenital heart disease (CHD) diagnosis during pregnancy) were more frequent in acquired cardiac group except for diagnosis of additional cardiac pathologies such as VHD or CHD during pregnancy.

On the contrary neonatal morbidities and outcomes were more frequent in the preexistent cardiac group.

In the multivariable model statistically significant association has been found between the adherence score and the following cardiac variables: the stronger association was with women with heart failure, than women in the preexistent cardiac group (p = 0.004), those women who had balloon valvoplasty (p = 0.004), those who required cardiac surgery (p = 0.041), and the investigations of CTPA inclusive of an echocardiogram (p = 0.047), whilst adjusting for all other covariates. No association with cardiac arrest.

Among obstetric variables elective LSCS mode of delivery had the statistically most robust association with an improved guideline adherence score (p < 0.001). Other associated obstetric variables were emergency LSCS (p = 0.0120), diagnosis of preeclampsia (p = 0.034) and placenta praevia complication (p = 0.011).

The only neonatal variable reporting a statistically significant association with the guideline

adherence score was NICU admission.

The study is very interesting for many reasons: first of all because cardiac pathology during pregnancy is a challenging field of increasing interest with few data and evidence reported in literature. The present study is well done and well argued nevertheless there are several points that deserve to be clarified.

MINOR ISSUES

1. In Discussion is enphasized that the association between the adherence score to the guidelines and several clinical variables indicate improved outcomes with greater adherence (line 315), but this statement must be better argued. It is implied that the outcomes are improved with greater GL adherence because the group of newly diagnosed women has a lower adherence to the guidelines and worse outcomes but it needs to be explained.

2. Since to be aware of the presence of heart disease and follow GL antenatal care is critical to evitate worse complications (ie maternal cardiac death, severe morbidity after cardiac arrest, Discussion line 319-346) is there any parameter in the score that can increase the sensitivity of the diagnosis of heart disease in the first trimester? Can you specify if are there GL indications on routine CV screening and if are considered in your score? (Discussion lines 359-369)

3. Discussion lines 389-397 needs to be clarified: The elective LSCS mode of delivery showed a statistically significant association with increased adherence to the guidelines. The rationale for elective LSCS included common obstetrical indications but it prompted early admission to hospital for cardiac optimisation In our cohort, the high rate of elective LSCS (although not considered the first option in thestatewide guidelines) but with increased guideline adherence scores related to other aspects of the guidelines yet improved the overall clinical outcome for both mother and baby. This result supports the safety of elective LSCS in severe and unstable cardiac conditions. What unstable cardiac conditions? What kind of cardiac optimization is done ?

4. In Methods (line131-134) needs to be clarified how is calculated “the total adherence score”, unless reporting table of the previous study, at least by mentioning the criteria by which it is computed. Moreover the sentence “The researchers agreed on the minimum acceptable score of 17 for the newly acquired cardiac group and 35 for the preexistent cardiac group”. Needs to be better explained.

5. In results when it comes to mean guideline adherence score for groups of patients is not clear how it is calculated (e.g at lines 284-285: Women with HF had a mean guideline adherence score of 3.546 units higher than those who did not (mean difference =3.546, 95% CI, (1.689, 5.403)).

REF

(1) Millington S, Arstall M, Dekker G, Magarey J, Clark R. Adherence to clinical practice guidelines for South Australian pregnant women with cardiac conditions between 2003 and 2013. PloS one. 2020;15(3)

6. PLOS authors have the option to publish the peer review history of their article (what does this mean?). If published, this will include your full peer review and any attached files.

Reviewer #1: **Yes: **Edison Muñoz Ortiz

Reviewer #2: No

Reviewer #3: No

---

## [Author Response · Author response to Decision Letter 0]

25 Jun 2021

Reviewer #1 

1. All statistics were performed under the supervision of a medical statistician who is also a co-author (Suzanne Edwards). The description of the model building completed in this paper is considered to be appropriate and rigorous based on Heinze and Dunkler 2017.

2. We have included a summary of our previous findings in the paper's introduction. We have not re-iterated the methods section of our last article as this is repetitive but referred to it. Therefore, like papers that have previously published their protocol and do not repeat the same information in the results paper. We hope this is satisfactory. 

Refer to pages 6-7, lines 119-132 of the clean copy of the manuscript. 

 3. The authors thank the reviewer for comments regarding the difference between the three hospitals. 

The authors agree that the difference between the three hospitals is important and was described in the previous publication. But this paper singular narrative aims to describe the frequencies of clinical variables, i.e. morbidities, outcomes and interventions inclusive of the cardiac groups of the women during pregnancy and their association with adherence scores. Therefore, the three hospitals were not included in the analysis for the following reasons. 

• The three tertiary hospitals identified in the study were in the same metropolitan city with access to the same recommended statewide perinatal guidelines for practice.

• The differences in real-life practice between the three hospitals were addressed in the 2020 publication; please refer to Millington SK et al. 2020 (20). The findings showed that across three SA Health public metropolitan, university-affiliated hospitals, the variance in adherence across the three hospitals correlated with the exposure to higher acuity cases. 

• As outlined in the same publication, there was cooperation between the three sites for interhospital transfer in high-risk cases. 

• Given the number of women in the cohort and the fact that the numbers were not equally matched for each hospital site, reducing the power to analyse, it was more feasible to look at the overall score for adherence in linear regression modelling and cardiac groups. 

• The disparities between the hospitals were not the aim of this paper and were highlighted in the previous publication. Refer to Page 6-7, lines 121 -132 clean copy of manuscript.

4. The authors thank the reviewer for requesting the description of outcomes and associated pathologies with the eleven cardiac arrest cases presented in Table VIII. This may be appropriate to include as a supplementary. Refer to Supplementary table VIII Cardiac Arrest cases, associated pathologies and outcomes.

5. Although the reviewer is concerned about the high percentage of LCSC rate in our cohort, the Australian Institute of Health and Welfare indicates that the LCSC rate for women giving birth for the first time from 2004 to 2018 in South Australia was 29.5% and overall in Australia was 30.1%. (https://www.aihw.gov.au/reports/per/095/ncmi-data-visualisations/contents/labour-birth/b5). The elective LCSC rate in our cohort of 261 was n= 74, 28.4%; however emergency LCSC rate was n=54, (20.7 %,) making our LCSC rate 49.1%. Solely cardiac indications for LCSC occurred in 20 women (7.6%), whereas obstetric or combined obstetric and cardiac indications occurred in the others. We found that the rationale for LCSC was frequently complex, and the decision was not based on a single parameter. This level of clinical complexity has been noted by previous registries (CAREPREG and ROPAC). 

In the CAREPREG study cohort of 546 women who completed pregnancies, 98% were live births, with 164 deliveries by caesarean section Most caesarean deliveries (96%) were for obstetric indications; maternal cardiac status was indicated in 4% as reported in Siu et al. 2001.

The ROPAC study with a focus on structural heart disease described two cohorts : 

PREG 1 cohort of 1,321 data collected before 2011 with reported LSCS 2,681 (46.7%) and emergency LSCS 766 (13.3%). 

PREG 2 cohort of 5,739 data collected from 2011 onwards with reported LSCS 2143(48.6%) and emergency LSCS 562 (12.7%). 

 Finally, in our cohort, the cardiac scenarios resulting in a decision for elective or emergency LCSC were (n=20) and those women who deviated from the delivery plan (n=13)

Valve disease with preeclampsia n=2

Valve disease with decompensated heart failure n=3

Stable but severe stenotic valve disease n=1

Dilated cardiomyopathy resulting is severe pulmonary hypertension and/or decompensated heart failure n=5

Worsening primary pulmonary hypertension with heart failure n=1

Marfans’ syndrome with significant aortic root dilatation n=3

Type B aortic dissection antepartum n=1

Complex congenital heart disease with preeclampsia n=1

cardiac arrest n=3

acute pulmonary oedema of unclear cause n=1

In general, the deviation of the plan from NVD to LCSC was related to a change in the clinical state with haemodynamic compromise or a new serious complication (cardiac, obstetric, neonatal or other medical reasons). We have categorised the indications for LCSC in Table IX, which may be an appropriate supplementary table.

6. We have included the number and reasons for deviation from the planned NVD to LCSC in supplementary table IX, so this should answer the reviewer's query.

7. The authors thank the reviewers for the feedback on acronyms. The manuscript has been checked for inconsistencies with abbreviations where they are not referenced before their use. Refer to page 5, lines 93, page 10 line 218, page 26, line 368, page 31 , line 488 and table 1 and supplementary tables have been updated.

Reviewer # 2

1. The authors thank the reviewer for their feedback on congenital neonatal conditions, particularly neonatal heart disease, and outcomes. The primary and secondary outcomes of neonatal mortality and morbidity are discussed earlier in the manuscript. Please refer to page 10.

"Preexistent maternal heart disease, possibly via a decline in maternal cardiac output during pregnancy (IUGR) and obstetric complications, are associated with an increased risk of neonatal complications (14, 19). Therefore, this study collected data for neonatal complications…." 

A small number (n=6) of neonates with congenital conditions diagnosed at birth or post-delivery did not influence overall adherence. All the babies survived, with some requiring further paediatric care and follow up, but the only neonatal variable with statistically significant association with guideline adherence score was NICU admission (P <0.001). Therefore, this data includes a small number of babies (n=6) diagnosed with congenital heart disease and abnormalities. Please refer to supplementary table S II on the neonatal diagnosis of congenital heart disease and abnormalities captured in the cohort with maternal cardiac status. The timing of the babies diagnosis occurred is made explicit for readers.

2. The authors thank the reviewers for feedback on what variables were included in the multivariable model. The manuscript outlined the statistical process in the manuscript with both univariate supplementary tables for transparency of the process see tables SV-SVII. In addition, the initial multivariable regression analysis for the separate cardiac, obstetric, and neonatal models are now presented in the results section table 2-4. However, the manuscript has been revised to improve clarity. Refer to pages 11-12 lines 243-250 and tables 2-4 in the clean manuscript. 

3. The authors thank the reviewers for feedback on the power analysis for outcomes of interest, given that many were rare events. The sample size calculation and power calculation was performed and reported in a previous publication based on adherence to guidelines, which is the primary research topic (20). Therefore sample size and power calculations are based on the primary hypothesis as reported in the previous paper. Clinical outcomes are a secondary topic. refer to page 8, lines 176-179.

4. The authors thank the reviewers for comment regarding the presentation of results. The two cardiac groups were not directly compared in this study. Still, they were included as binary variables in the cardiac complications /interventions regression model, where the outcome was the total adherence score. The comparison of total adherence score in the preexistent cardiac group was found to be significant ( p = 0.004 ) Table 5. 

Reviewer#3 

 1. The authors thank the reviewers for the feedback on the opening statement in the discussion section, referenced with Table 2, where the final multivariable linear regression model results are presented. Accordingly, the manuscript revised to reflect the findings and statement. 

This comprehensive retrospective study supports the hypothesis of an association between the adherence score to the guidelines and some clinical variables indicating improved outcomes with greater adherence (Table 2). But the most significant was decompensated heart failure (p < 0.001) and elective LSCS elective (p <0.001) contingent upon the inclusion of multidisciplinary team collaboration in high risk pregnancies, therefore, increased adherence as previously reported (20).

 Further discussion is included in the manuscript on heart failure and delivery modes pp 29-30. Refer to page 26, lines 370-374, page 29, lines 442-478.

2. We thank the reviewers for the feedback question on which parameters in the score are more likely to increase sensitivity to heart disease diagnosis in the first trimester?

No single parameter increases sensitivity to diagnosis of heart disease; however, clinicians' awareness and preparedness of women's preexisting cardiac conditions and the guideline appropriate care helps anticipate potential pregnancy complications. Therefore clinical physical assessment and accurate history taking in the first trimester are strongly recommended. Likewise, in women with acquired cardiac conditions, the health care professional's clinical intuitiveness to adjust the care trajectory as per the guidelines may mitigate severe complications—discussed in manuscript page 28. 

The clinical audit examined routine CV screening in the overall adherence score, the healthcare professional who assessed the woman at the first visit. Still, the healthcare 

professional's expertise and knowledge who assessed the pregnant women at the first antenatal visit influenced the information captured. 

Routine prenatal health (cardiac) screening is not undertaken on all pregnant women. In South Australia over the last twenty years, midwives triage booking visits, which has worked well for most women and has improved detection of psychosocial issues and domestic violence. But there is no medical input. Therefore depending on the expertise and knowledge of midwives of risk factors, i.e. IV drug user, indigenous and RHD, cardiac /family history might be missed. Importantly, the perinatal guidelines for women with cardiac disease in pregnancy recommend identifying women with previous pregnancies and cardiac history, cardiac assessment, preconception cardiac education where applicable, i.e. preexistent cardiac group and specific cardiac investigations and neonatal risk assessment pertinent to cardiac conditions. This information was collected in the clinical audit, scored and reported in the logistic regression analysis in Millington et al. (2020).

 3. The information in Supplementary Table S IX should help to clarify the reviewer's concerns. Cardiac optimisation included advanced life support in cardiac arrest, diuresis in decompensated heart failure, respiratory support to manage hypoxia, inotropic support to manage cardiogenic shock, haemodynamic monitoring during and immediately after delivery and management of cardiac arrhythmias with anti-arrhythmic medication, DC cardioversion or implantation of an internal cardiac electrical device. We have included this in the discussion to assist in the clarification as requested. refer to pages 30 , Lines 446-471 plus supplementary Table IX.

4. The authors thank the reviewers for further clarification on calculating the total adherence score and minimum acceptable score for the two cardiac groups. Calculation of total score was reported in earlier publication and achieved by quantifying perinatal guidelines, i.e. planned care variables were used to calculate the total score of 40 (N=40). The manuscript has been revised as requested. Page 7, lines 142 – 151.

5. The authors thank the reviewers for further clarification on guideline adherence score for groups of patients is calculated. The final Multivariable linear regression results Table 2 includes the unstandardised coefficient beta, and so the mean difference is the calculation from the unstandardised beta coefficient linear regression equation. See page 19, lines 317-319. 

References

 Heinze, G., & Dunkler, D. (2017). Five myths about variable selection. Transplant International, 30(1), 6-10. doi:https://doi.org/10.1111/tri.12895

Millington, S., Arstall, M., Dekker, G., Magarey, J., & Clark, R. (2020). Adherence to clinical practice guidelines for South Australian pregnant women with cardiac conditions between 2003 and 2013. PLoS One, 15(3), e0230459-e0230459. doi:10.1371/journal.pone.0230459.

Millington S, Magarey J, Dekker GA, Clark RA. Cardiac conditions in pregnancy and the role of midwives: A discussion paper. Nursing Open. 2019 0(July 23). doi: 10.1002/nop2.269.

Siu, S. C., Marcotte, F., Taylor, D. A., Gordon, E. P., Spears, J. C., Tam, J. W., . . . Investigators, C. (2001). Prospective multicenter study of pregnancy outcomes in women with heart disease. CIRCULATION, 104(5), 515-521. doi:10.1161/hc3001.09343.

Silversides, C. K., Grewal, J., Mason, J., Sermer, M., Kiess, M., Rychel, V. . . . Siu, S. C. (2018). Pregnancy Outcomes in Women With Heart Disease: The CARPREG II Study. Journal of the American College of Cardiology, 71(21), 2419-2430. https://doi.org/10.1016/j.jacc.2018.02.076.

Roos-Hesselink, J., Baris, L., Johnson, M., De Backer, J., Otto, C., Marelli, A. . . . Hall, R. (2019). Pregnancy outcomes in women with cardiovascular disease: evolving trends over 10 years in the ESC Registry of Pregnancy and Cardiac disease (ROPAC). European Heart Journal, 00, 1-8. doi:10.1093/eurheartj/ehz136.

---

## [Decision Letter · Decision Letter 1]

9 Jul 2021

The association between guidelines adherence and clinical outcomes during pregnancy in a cohort of women with cardiac co-morbidities.

PONE-D-21-08006R1

Dear Dr. Millington,

We’re pleased to inform you that your manuscript has been judged scientifically suitable for publication and will be formally accepted for publication once it meets all outstanding technical requirements.

Kind regards,

Sara Ornaghi, M.D., Ph.D.

Academic Editor

PLOS ONE

Additional Editor Comments (optional):

Reviewers' comments:

Reviewer's Responses to Questions

**Comments to the Author**

1. If the authors have adequately addressed your comments raised in a previous round of review and you feel that this manuscript is now acceptable for publication, you may indicate that here to bypass the “Comments to the Author” section, enter your conflict of interest statement in the “Confidential to Editor” section, and submit your "Accept" recommendation.

Reviewer #1: All comments have been addressed

Reviewer #3: All comments have been addressed

2. Is the manuscript technically sound, and do the data support the conclusions?

Reviewer #1: Yes

Reviewer #3: Yes

3. Has the statistical analysis been performed appropriately and rigorously? 

Reviewer #1: N/A

Reviewer #3: Yes

4. Have the authors made all data underlying the findings in their manuscript fully available?

Reviewer #1: Yes

Reviewer #3: Yes

5. Is the manuscript presented in an intelligible fashion and written in standard English?

Reviewer #1: Yes

Reviewer #3: Yes

6. Review Comments to the Author

Reviewer #1: (No Response)

Reviewer #3: The Authors adequately addressed my comments clarifying statements and including data also in supplementary tables.

7. PLOS authors have the option to publish the peer review history of their article (what does this mean?). If published, this will include your full peer review and any attached files.

Reviewer #1: No

Reviewer #3: No

---

## [Editor Report · Acceptance letter]

16 Jul 2021

PONE-D-21-08006R1 

The association between guidelines adherence and clinical outcomes during pregnancy in a cohort of women with cardiac co-morbidities. 

Dear Dr. Millington:

I'm pleased to inform you that your manuscript has been deemed suitable for publication in PLOS ONE. Congratulations! Your manuscript is now with our production department. 

Kind regards, 

on behalf of

Dr. Sara Ornaghi 

Academic Editor

PLOS ONE